# Diagnosis and Treatment of Obstructive Hypertrophic Cardiomyopathy

**Gaetano Todde, Grazia Canciello, Felice Borrelli, Errico Federico Perillo, Giovanni Esposito, Raffaella Lombardi and Maria Angela Losi ***

Department of Advanced Biomedical Sciences, Federico II University, 80138 Naples, Italy
* Correspondence: losi@unina.it; Tel.: +39-081-726-22-21

**Abstract:** Left ventricular outflow obstruction (LVOTO) and diastolic dysfunction are the main pathophysiological characteristics of hypertrophic cardiomyopathy (HCM)LVOTO, may be identified in more than half of HCM patients and represents an important determinant of symptoms and a predictor of worse prognosis. This review aims to clarify the LVOTO mechanism in, diagnosis of, and therapeutic strategies for patients with obstructive HCM.

**Keywords:** hypertrophic cardiomyopathy; left ventricular outflow obstruction; systolic anterior motion

## 1. Introduction

Hypertrophic cardiomyopathy (HCM) is defined by an increased left ventricular (LV) wall thickness that is not only explainable by abnormal loading conditions [1,2]. LV hypertrophy in the absence of cardiovascular diseases occurs in approximately 1:500 subjects in the general population [3]; when both clinical and genetic diagnoses are considered, this prevalence increases to 1 case per 200 [3].

The pathophysiology of HCM is characterized by diastolic dysfunction and left ventricular outflow obstruction (LVOTO). LVOTO is characteristically dynamic and may be identified in about two thirds of HCM patients, with one-third of patients presenting with LVOTO during rest and the other one-third with latent LVOTO being elicited only during a Valsalva maneuver or exercise [4].

It has been shown that LVOTO during rest is a strong, independent predictor of progression toward severe heart failure symptoms and death in patients with HCM [5]. Moreover, gender differences have been reported in the role of LOVTO in the course of HCM; in fact, the presence of LVOTO has been associated with an increased risk of symptom progression or death due to heart failure in women versus men [6].

According to the results of genetic screening, after excluding HCM phenocopies (such as those of the Fabry-Anderson disease and amyloidosis), HCM patients may be divided into two subgroups, those carrying and those not carrying a sarcomere gene mutation (sarcomeric-positive and sarcomeric-negative HCM, respectively) [7]. LVOTO was initially considered typical of sarcomeric-positive patients, but over time it has been observed that sarcomeric-negative patients show a higher prevalence of LVOTO [8].

In this review, we will discuss the mechanisms in, diagnosis of and therapeutic strategies for patients with obstructive HCM (HOCM).

## 2. Mechanism of Intraventricular Obstruction

In HCM obstruction may occur at the LVOT or at the midventricular level. Both types of obstruction may occur alone or in combination in the same HCM patient (Figure 1).

The main components involved in the generation of LVOTO are the interventricular septum (IVS), the fibrous trigones and subaortic curtain, the mitral valve and the subvalvular mitral apparatus [9].

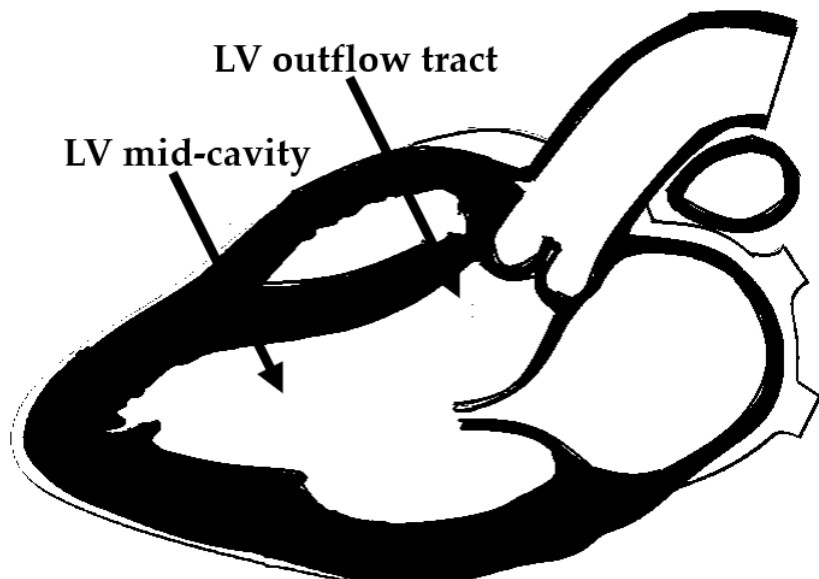

**Figure 1.** Localization of intraventricular obstruction in hypertrophic cardiomyopathy. LV = left ventricular.

The hypertrophic process in HCM preferentially affects the IVS (Figure 1). IVS thickness may vary from 15 to 50 mm in extreme cases. Hypertrophy of the muscular septum, by leading to a reduction in the septo-mitral angle at the level of the fibrous trigones, is responsible for the acceleration of the systolic flow. In rare cases, massive IVS hypertrophy may even result in direction changes of the systolic flow.

IVS and a narrowed opening angle at the level of the trigones create a disturbed pattern of flow through this area which is responsible over time for a fibrotic reaction at the level of the fibrous trigones and subaortic curtain. This fibrotic process may significantly reduce the mobility and excursion of trigones, contributing to the dynamic process of LVOTO.

Several structural and functional abnormalities of the mitral valve may substantially contribute to LVOTO. The increase in length of the anterior and/or posterior mitral leaflets [10,11] (Figure 2), forward displacement of the anterior papillary muscle, and laxity of the mitral valve chords (Figure 2) have been implicated in the anterior displacement of the coaptation line and in abnormal leaflet coaptation, characterized by a posterior leaflet coapting the mid-portion of an anterior leaflet, leaving its distal portion unsupported. This part of the leaflet causes dynamic obstruction, via a combination of suction, caused by the Venturi effect of the rapid flow in LVOT, and a dragging mechanism, caused by blood pushing the leaflet towards the septum (Figure 2).

These dynamic changes result in mitral systolic anterior motion (SAM), which contributes to both LVOTO and mitral regurgitation. SAM is a common but non-specific finding in HCM. In fact, alternative causes of SAM, independently of the presence of LV hypertrophy, are hypovolemia, inotropic drug use, small ventricles in normal or hypertensive individuals, and mitral valve surgical repair.

Mid-cavity obstruction (MCO) is caused by marked mid-septal hypertrophy resulting in contact between the IVS and a hypercontractile LV-free wall or an anomalous papillary muscle during contraction. Approximately 10% of HCM patients have MCO, so it represents an independent predictor of adverse outcomes.

There is a strong association between MCO and apical aneurysm formation. An apical aneurysm is a thin-walled dyskinetic or akinetic segment of the most distal portion of the LV chamber with relatively wide communication with the LV cavity. Its presence has been associated with an increased risk of an arrhythmic and thromboembolic event and sudden death [12].

HCM patients with MCO and apical aneurysms may not show high mid-LV velocities at the Doppler evaluation because of a complete closure of the mid-LV areas which may look like a sphincter closing aneurysm In this case, the it is possible to observe a paradoxical

blood flow emerging from the apex toward the base of the LV during a diastole when the sphincter is released [13].

For this reason, patients with MCO and an apical aneurysm may be mistakenly classified as "nonobstructive" despite the presence of severe obstruction [13].

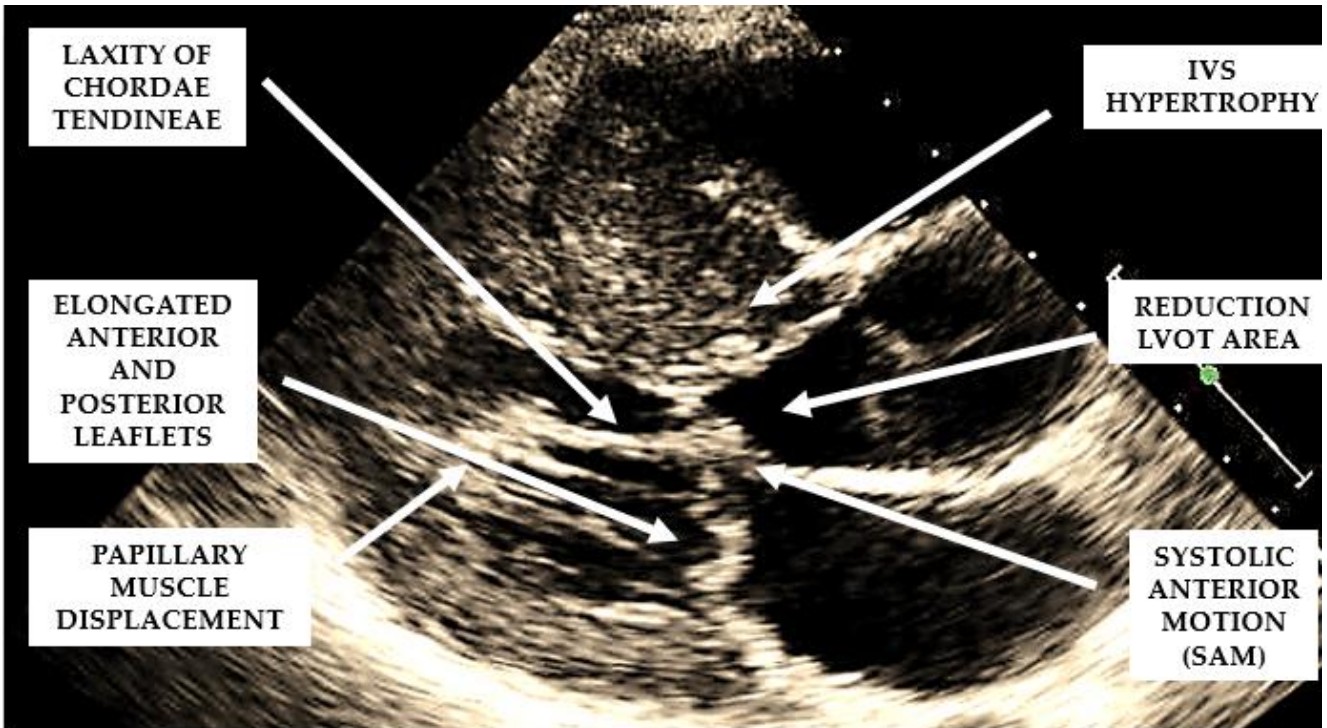

**Figure 2.** Mechanisms involved in the generation of left ventricular outflow tract obstruction in patients with hypertrophic cardiomyopathy.

## 3. Diagnosis

### 3.1. Echocardiography

Echocardiography plays a central role in the diagnosis and monitoring of HOCM. During the execution of an echocardiogram, particular attention should be paid to the M-mode visualization of the mitral valve, the 2D evaluation of LV morphology, the color Doppler appearance of the intracavity flows from the apical view, and to the detection and measurement of LVOT and mid-LV gradients by Doppler, both at rest and during a Valsalva maneuver.

#### 3.1.1. M-MODE Echocardiography

SAM and its severity are assessed via the M-mode from a short-axis view at the mitral valve level. SAM is defined as (1) incomplete if the mitral anterior and/or posterior leaflets do not touch the IVS, as (2) mild if mitral–septal contact occurs in the late systole and for less than 10% of the duration of the systole, and as (3) severe if it starts mid-systole and occupies more than 30% of its duration.

SAM is characterized by an abrupt anterior movement of the mitral valve during contraction, reaching its peak before the maximum movement of the posterior wall; this characteristic allows the differentiation of true SAM from SAM produced by an exaggerated anterior motion of the mitral valve, which reaches its peak after the full contraction of the posterior wall, i.e., "pseudo-SAM" [14] (Figure 3).

It has been demonstrated that there is a positive correlation between the severity of SAM and the severity of LVOTO evaluated invasively and that a reliable non-invasive method with which to estimate the LVOT gradient may be obtained with the following formula: $[(x/y) \times 25] + 25$, where x is the duration of SAM–septal contact and y is the interval

from the beginning of SAM to the occurrence of SAM–septal contact (x) (Figure 4) [15,16]. This formula may not be used to calculate the MCO.

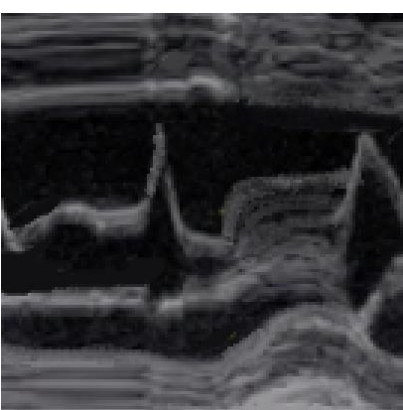

**Figure 3.** Pseudo SAM: the peak of SAM reaches after the full contraction of the posterior wall.

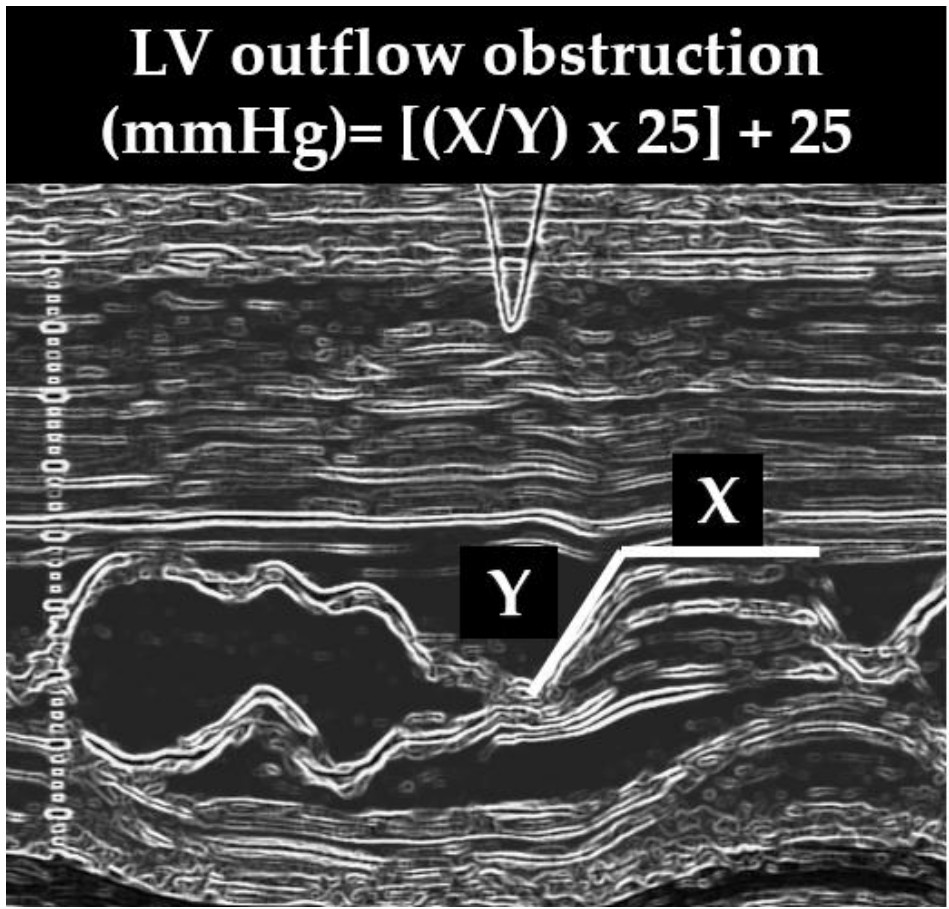

**Figure 4.** Severe systolic anterior motion (SAM). X = duration of SAM, Y = time elapsed between the beginning of SAM and the occurrence of SAM–septal contact. LV = left ventricular.

### 3.1.2. 2-D Echocardiography

When evaluating the presence and severity of obstruction in HCM, a systematic assessment of all the components of the mitral valve apparatus via 2D echocardiography is required. This technique allows a visualization of the presence and distribution of LV hypertrophy, the presence of SAM, elongation of mitral valve leaflets, displacement of papillary muscles, laxity of tendon cords and LVOT diameter reduction (Figure 2). In

addition, 2D echocardiography allows a diagnosis of MCO via the observation of a typical hourglass appearance of the LV due to systolic septal contact with the anterolateral wall, which induces sphincter-like cavity obliteration, creating two distinct (basal and apical) LV chambers (Figure 5). In addition, 2D echocardiography allows a visualization of the presence of an apical aneurysm and any thrombotic formation. In this setting, contrast echo may also be helpful for the correct diagnosis.

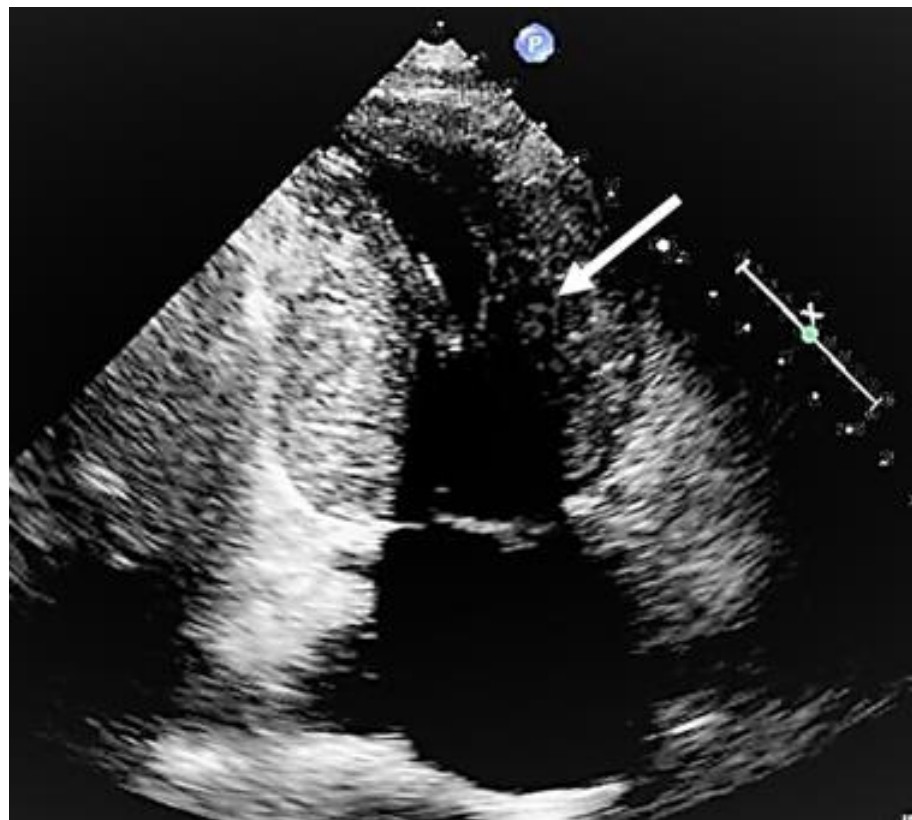

**Figure 5.** Mid-cavity obstruction; typical hourglass appearance of left ventricle (arrow).

When the quality of the transthoracic echo images is not optimal, a transesophageal echo may be considered.

### 3.1.3. Color Doppler Echocardiography

Color Doppler echocardiography identifies the presence and degree of mitral regurgitation and helps to understand the level at which obstruction occurs.

Mitral regurgitation (MR), caused by SAM, is observed in almost all patients with HOCM anterior mitral leaflet elongation and the associated increased mobility impairs adequate leaflet coaptation, resulting in SAM-related, eccentric posterior and lateral MR (Figure 6). When additional mitral valve abnormalities other than SAM are not observed, a direct relation between the pressure gradient and the severity of MR is evident. A central or anterior jet often indicates the presence of organic mitral valve disease [17].

In patients with MCO and an apical aneurysm, color Doppler echocardiography is prone to aliasing in the apical area due to the vortexes generated by the blood sequestered inside the aneurysm.

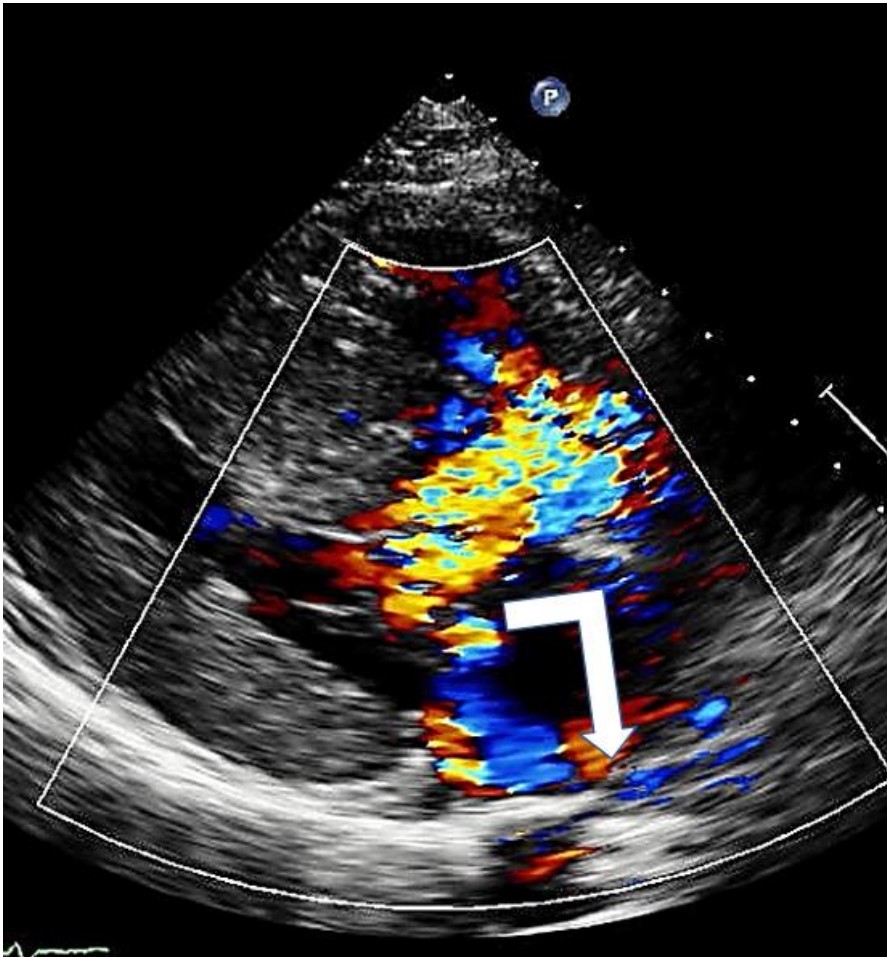

**Figure 6.** Mitral regurgitation in hypertrophic cardiomyopathy due to systolic anterior motion of the mitral valve; typical eccentric posterior direction (arrow).

### 3.1.4. Continuous and Pulsed Doppler Echocardiography

HOCM is defined by a peak Doppler LVOT pressure gradient of ≥30 mm Hg at rest, or during physiological provocation such as that involved in a Valsalva maneuver, standing or exercise [1,18]. Exercise echocardiography using treadmill exercise is an important technique in the detection of inducible HOCM [17]; it should be performed in symptomatic patients if Valsalva maneuvers fail to induce LVOTO of up to ≥50 mmHg.

Continuous Doppler from the apical views is the most reliable method with which to measure intraventricular gradients, whereas pulsed Doppler is used to locate the exact point where obstruction starts, since obstruction may occur at different levels inside the LV cavity.

It is therefore advisable to use pulsed Doppler to map the flow from the LV apex to the LVOT so as not to miss intraventricular areas of increased gradients (Figure 7).

The typical morphological appearance of the Doppler signal caused by a LVOTO is a "dagger-shaped" and late-peaking curve (Figure 8A) while the mitral regurgitation shape upon continuous Doppler is parabolic (Figure 8B).

Care should be taken to avoid contamination with the MR jet, which may cause an overestimation of the obstruction (Figures 8–10).

The MCO spectral Doppler profile has a similar profile to that of LVOTO; however, some little differences may help in the correct identification of these two types of obstruction. LVOT acceleration is initially slow, followed by a second phase of acceleration, whereas in MCO, the second phase of acceleration appears to be steeper and faster. Indeed, the second phase of acceleration of the intraventricular gradient's spectral profile appears to be almost

exponential and can be compared, rather than to a "dagger", to one side of an inverted half-pipe skateboard ramp [19].

In the presence of an apical aneurysm, a paradoxical apex to the base diastolic gradient [20] has been described.

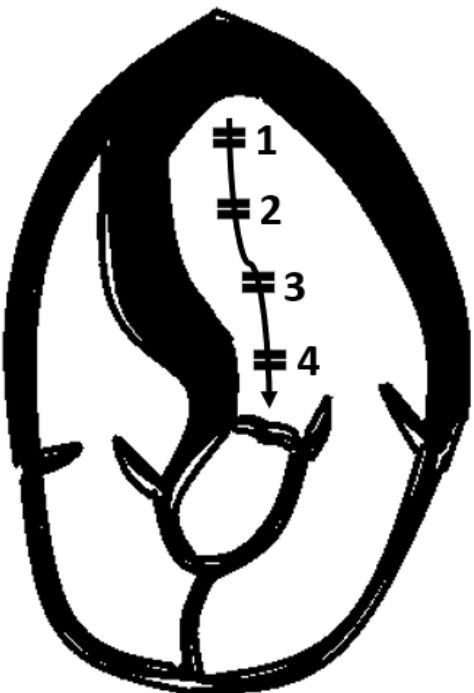

**Figure 7.** To investigate intraventricular gradient, it is recommended to move pulsed Doppler from apex (1) to left ventricular outflow tract (4).

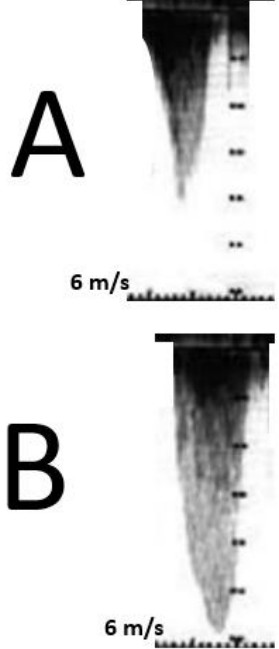

**Figure 8.** Shows the contemporary record of mitral regurgitation, LVOTO and MCO.

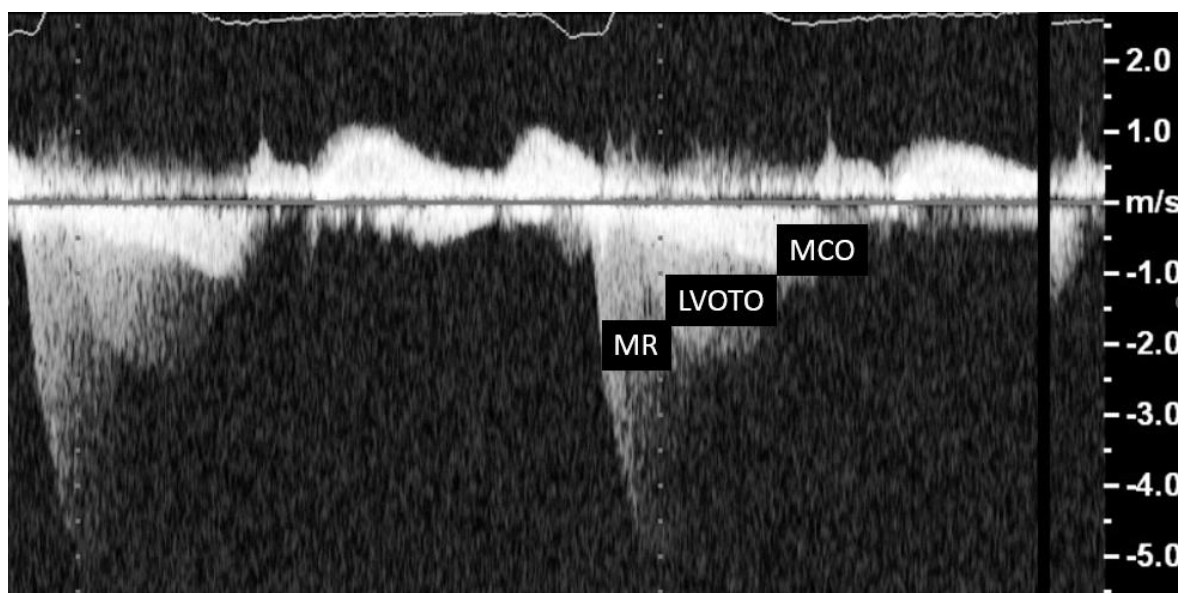

**Figure 9.** Left ventricular outflow tract obstruction (LVOTO); contemporary acceleration due to mid cavity obliteration (MCO) and contemporary record of mitral regurgitation (MR) curves with continuous-wave Doppler.

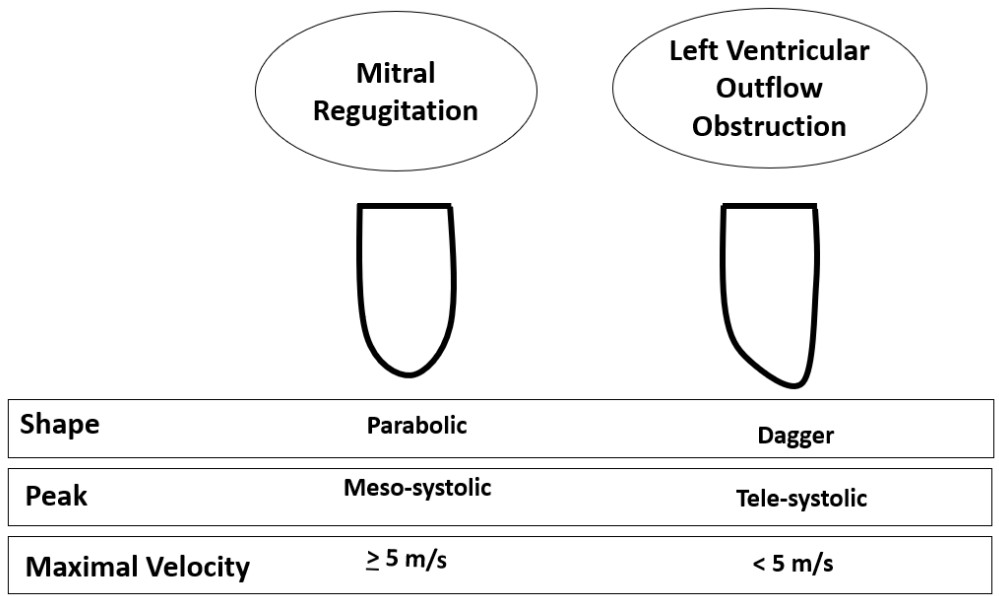

| | Mitral Regurgitation | Left Ventricular Outflow Obstruction |
|---|---|---|
| **Shape** | Parabolic | Dagger |
| **Peak** | Meso-systolic | Tele-systolic |
| **Maximal Velocity** | ≥ 5 m/s | < 5 m/s |

**Figure 10.** Differences in continuous Doppler waves between mitral regurgitation and left ventricular outflow tract gradient.

### 3.2. Cardiac Magnetic Resonance

Cardiac magnetic resonance (CMR) protocols in HCM should always include an assessment of the mitral valve, with slices positioned perpendicular to the valve plane (through-plane) along with in-plane views of the valve orifice. Velocity-encoded CMR imaging is added to determine the peak velocity of blood flow through the left ventricle [21].

CMR is superior to standard 2D echocardiography in the detection of LV apical and anterolateral hypertrophy, apical aneurysms, thrombi, myocardial crypts, and papillary muscle abnormalities; however, small structures, such as the mitral valve apparatus, are not well-visualized because imaging slices in CMR are thick and potentially lead to partial volume artifacts (Figure 11).

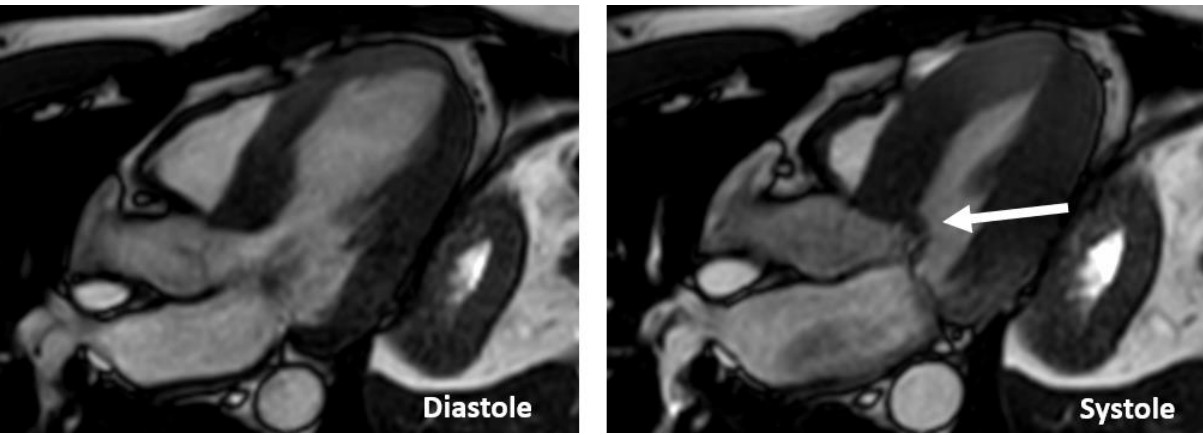

**Figure 11.** Images of long-axis CMR view of a patients with HCM showing SAM (arrow).

Accurate quantification of LVOTO is time-consuming, prone to error and can only be measured at rest. For these reasons, Doppler echocardiography is the modality of choice for the quantification of LVOTO. In addition, CMR with gadolinium, via the distribution analysis of late enhancement, helps in the differential diagnosis of HCM with HCM phenocopies such as those of amyloidosis and Fabry disease (Figure 12).

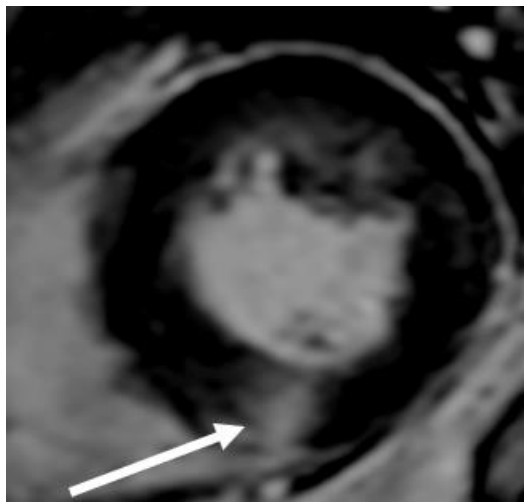

**Figure 12.** Late enhancement at insertion points of the right ventricular wall into the posterior interventricular septum, as is typical of hypertrophic cardiomyopathy (arrow).

*3.3. Cardiac Computed Tomography*

Cardiac computed tomography (CCT) provides accurate measurements of LV wall thickness, and a delineation of LV volumes, mass, and ejection fraction; overall these measurements have a good correlation with those obtained by CMR. In addition, coronary arteries and cardiac valves are well-assessed by CCT. For this reason the European Society of Cardiology Guidelines on HCM recommends that CCT should be considered in patients with poor-quality echocardiographic images or with contraindications to performing a CMR.

**4. Therapy**

Treatment of LVOTO is indicated in patients with lifestyle-limiting symptoms only. Negative inotropic and chronotropic medications are indicated as the first-line therapy. If patients remain symptomatic, or remain in the presence of side effects, surgery is suggested only when performed in experienced centers. Percutaneous septal ablation is a potential

alternative to myectomy for patients with elevated surgical risk and is compatible with the mechanisms of LVOTO and an optimal coronary anatomy.

### 4.1. Medical Therapy

Pharmacological treatment represents the first line of the management of symptomatic LVOTO. Medical therapy has been shown to efficiently control symptoms in 65% of HOCM patients avoiding an invasive septal reduction intervention (Figure 13).

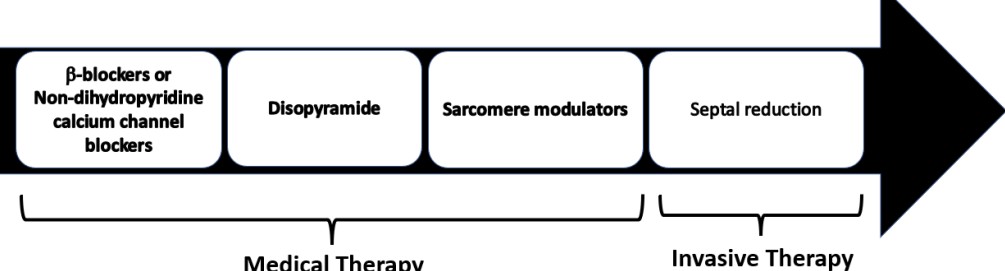

**Figure 13.** Management of symptomatic HCM patients with LVOTO. Non-vasodilating β-blockers are recognized as the first-line therapy, with non-dihydropyridine calcium channel blockers as the alternative in patients with contraindications to β-blockers.

The beneficial effects of b-blockers are imparted through the prevention of catecholaminergic increases in heart rate, in ventricular contractility and atrio-ventricular conduction. The collective effects of b-blockers lead to improved ventricular relaxation, increased diastolic filling time, reduction in LV end-diastolic pressure and improved perfusion [22]. Furthermore, b-blockers are used in HCM to control the arrhythmic burden in patients with frequent supraventricular or ventricular ectopies, and for rate control in patients with atrial fibrillation. Propranolol was the first b-blocker used and in old studies showed a significant benefit for symptoms associated with a significant reduction in LVOTO [23–27]. More selective beta blockers such as nadolol and bisoprolol are equally effective in controlling exercise induced LVOTO [28].

In a recent double-blind randomized crossover trial, compared with the placebo, metoprolol demonstrated a LVOTO reduction at rest and during exercise, provided symptom relief, and improved quality of life in patients with HOCM. However, in this trial, despite the improvement in symptoms and reduction in LVOTO, maximum exercise capacity and oxygen consumption (VO2) remained unchanged, probably because the treatment reduced the heart rate only by 25%, a decline that probably was not fully compensated for via an increase in stroke volume, even with a prolonged diastolic filling time. [29].

Non-dihydropyridine calcium channel blockers, such as verapamil and diltiazem, can be used when β-blockers are contraindicated or ineffective, but close monitoring is required in patients with severe obstruction (LVOTG ≥ 100 mm Hg) or elevated pulmonary artery systolic pressures, as these drugs may provoke pulmonary edema [30–33].

Disopyramide (antiarrhythmic class IA agent), due to its negative inotropic effect, is effective at reducing LVOTO. This drug is usually used in combination with β-blockers because it increases the velocity of AV conduction and consequently the ventricular rate. The safety and efficacy of disopyramide were demonstrated in a large multicenter registry [34]. Adverse drug reactions include QT prolongation and anticholinergic effects (xerostomia, nausea, constipation, and urinary retention). An electrocardiogram should be performed before and after initiation of the drug, to evaluate the corrected QT (QTc) interval. It is essential to inform patients of the need to avoid concomitant therapy with other drugs associated with QTc prolongation; conditions that favor dehydration or electrolyte imbalance should also be avoided. Patients with MCO should be treated with high-dose β-blockers, verapamil or diltiazem, but medical therapy in this setting is often not useful (Table 1).

**Table 1.** Commonly used drugs in obstructive hypertrophic cardiomyopathy.

| Drug | Indication | Side Effects |
|---|---|---|
| Propranolol | Short half-life. Preferred in infants. | Chronotropic incompetence; broncho stenosis; hypotension; AV conduction decrease. |
| Nadolol | First-line therapy. | Chronotropic incompetence; broncho stenosis; hypotension; AV conduction decrease. |
| Atenolol | First-line therapy. | Chronotropic incompetence; broncho stenosis; hypotension; AV conduction decrease |
| Bisoprolol | Not usually used for vasodilatory effect. | Chronotropic incompetence; broncho stenosis; hypotension; AV conduction decrease. |
| Metoprolol | Short half-life; First-line therapy. | Chronotropic incompetence; broncho stenosis; hypotension; AV conduction decrease. |
| Verapamil | Alternative to b-blockers in patients without left ventricular disfunction, high LVOTO and PAPs | AV conduction decrease; ankle edema |
| Diltiazem | Little evidence available | AV conduction decrease; ankle edema |
| Disopyramide | Second-line therapy in association with first-line therapy | QTc prolongation; anticholinergic effects |
| Mavacamten | Not yet available in Europe. Used in no-response medical therapy patients | Possible reduction in global systolic left ventricular function |

*4.2. Novel Therapies*

Recently, several new drugs have been investigated in HOCM. A hypercontractile state emerged as a suitable target for the development of a novel pharmacological disease-specific approach. In 2016, mavacamten (MYK-461), a small molecule that reduces the contractility of cardiac myocytes by inhibiting the ATPase activity of myosin, was reported to be effective at attenuating, and even reversing the key phenotypic aspects of HCM in a transgenic mouse model [35]. Afterwards, in a Phase 2 pilot study involving 21 symptomatic patients with HOCM, mavacamten was generally well-tolerated and significantly reduced resting and post-exercise peak LVOTO [36].

Based on these results, the EXPLORER-HCM trial, a multicenter, Phase 3, randomized, double-blind, placebo-controlled study, evaluated the efficacy and safety of mavacamten in adults with symptomatic HOCM. The EXPLORER-HCM trial included 251 patients (age $58 \pm 5$ years; 41% women) with HOCM and NYHA class II or III, randomized at 1:1 to receive, once daily, oral mavacamten, or a matching placebo for 30 weeks. The blind oral dose titration (2.5, 5, 10 or 15 mg) was individualized to achieve target a reduction in LVOTO to less than 30 mmHg and a mavacamten plasma concentration between 350 ng/mL and 700 ng/mL. Over 90% of patients included in the trial were already on a treatment with β-blockers or calcium antagonists. A significant proportion of patients on mavacamten (37%) met the primary endpoint ($\geq 1.5$ mL/kg per minute increase in peak oxygen consumption and at least one NYHA class reduction, or $\geq 3.0$ mL/kg per minute increase in peak oxygen consumption without NYHA class worsening) compared with the 22% who did so on the placebo. In addition, patients on mavacamten showed a greater reduction in LVOTO

after exercise, a greater increase in peak oxygen consumption, and symptom improvement compared with those on placebo [37].

In EXPLORER-HCM substudies, over the 30-week treatment period, mavacamten showed significant improvements in several echocardiographic parameters, such as reductions in LV wall thickness and mass, increases in LV cavity dimensions, reductions in left atrial volumes and improvements of diastolic parameters including E/e' [38].

VALOR-HCM is a multicenter Phase 3, double-blind, placebo-controlled, randomized study. The study population consisted of approximately 100 patients (≥18 years old) with symptomatic HOCM who were considered eligible to septal reduction therapy. The goal of the trial was to assess the safety and efficacy of adding mavacamten to maximally tolerated medical therapy among patients with HOCM. The results of this trial suggested that mavacamten improved symptoms and significantly reduced eligibility for septal reduction therapy among symptomatic patients with HOCM [39].

The potential inconveniences of mavacamten treatment are the six-week period required to reach a steady-state concentration, and the induction of the cytochromes P450 3A4 (CYP3A4) and 2B6 (CYP2B6) shown in in vitro studies suggesting possible pharmacological interactions with other drugs which are metabolized trough this cytochrome system.

In 2022, the US Food and Drug Administration (FDA) approved mavacamten capsules for treating adults with symptomatic NYHA class II–III HOCM to improve exercise capacity and symptoms and will be available soon also in Europe.

Aficamten (CK-274) is a novel cardiac myosin inhibitor which was recently released. Aficamten has a half-life adequate for single daily administration, achieving a steady state within 2 weeks (so faster than mavacamten), and with no evidence of cytochrome P450 induction or inhibition. For the pharmacokinetic characteristics, this drug may be considered a more manageable sarcomere modulator [40].

*4.3. Invasive Septal Reduction Therapies*

Septal reduction therapies include septal myectomy (SM) and alcohol septal ablation (ASA). This invasive technique should be considered in patients with an LVOTO gradient of ≥50 mm Hg, moderate-to-severe symptoms (NYHA Class III–IV) and/or recurrent exertional syncope despite having maximally tolerated medical therapy. These procedures should be performed in experienced centers with a multidisciplinary team of experts, as indicated by American and European guidelines [1,2].

SM is most commonly performed through the Morrow technique, in which through two side by side myotomies, a rectangular trough is created in the basal septum below the aortic valve. The first myotomy is made with an angled handle knife just to the right of the commissure between the left and right coronary leaflets. The blade is inserted into the septum, in the long axis of the ventricle for 4 cm and is removed with a sawing motion directed toward the ventricular lumen and the retractor. A second myotomy is made about 1 cm to the right (clockwise) of the first myotomy. The incisions are then deepened, if necessary. The myotomies are usually 12–15 mm in depth at in most prominent region of the septum. A transverse incision is then made at the base of the valve leaflet connecting the proximal portions of the two myotomies [41].

As pointed out previously, HCM frequently presents with several anatomic alterations of the mitral valve apparatus and these structural abnormalities may predispose one to residual SAM and result in a suboptimal outcome with a persistence of outflow obstruction and mitral regurgitation after surgery [42].

In patients with marked mitral leaflet elongation and/or moderate-to-severe mitral regurgitation, septal myectomy can be combined with one of several adjunctive procedures, including mitral valve replacement, posterior-superior realignment of the papillary muscles, partial excision and mobilization of the papillary muscles, anterior mitral leaflet plication, and anterior leaflet extension using a glutaraldehyde-treated pericardial patch that stiffens the mid-portion of the leaflet. When there is a co-existing mid-cavity obstruction, the

standard myectomy can be extended distally into the mid-ventricular region around the base of the papillary muscles.

The main surgical complications of SM are AV nodal block, ventricular septal defects, and aortic regurgitation (AR), but these complications are uncommon in experienced centers. Surgical mortality is around 3–4%.

ASA consists of a selective infusion of high-grade alcohol into a septal branch supplying the basal interventricular septum to create an iatrogenic localized scar with the aim of LVOTO reduction. This procedure is less invasive than surgical myectomy is, and requires a shorter hospital stay.

The main complications of ASA are AV block, which may occurs in 7–20% of patients, and ventricular arrhythmia due to reentry caused by the scar, while the procedural mortality is similar to that of an isolated myectomy [43,44].

Several meta-analyses suggest that there are no differences in short- and long-term all-cause mortality, cardiovascular mortality and sudden cardiac death between ASA and SM. In alcohol ablation, peri-procedural complications are less common but re-intervention and pacemaker implantation are more common. Long-term symptomatic improvement and LVOT gradient reduction favor SM over ASA [45,46] (Figure 14). This makes SM the first-line therapy in young patients without comorbidities and with low surgical risks.

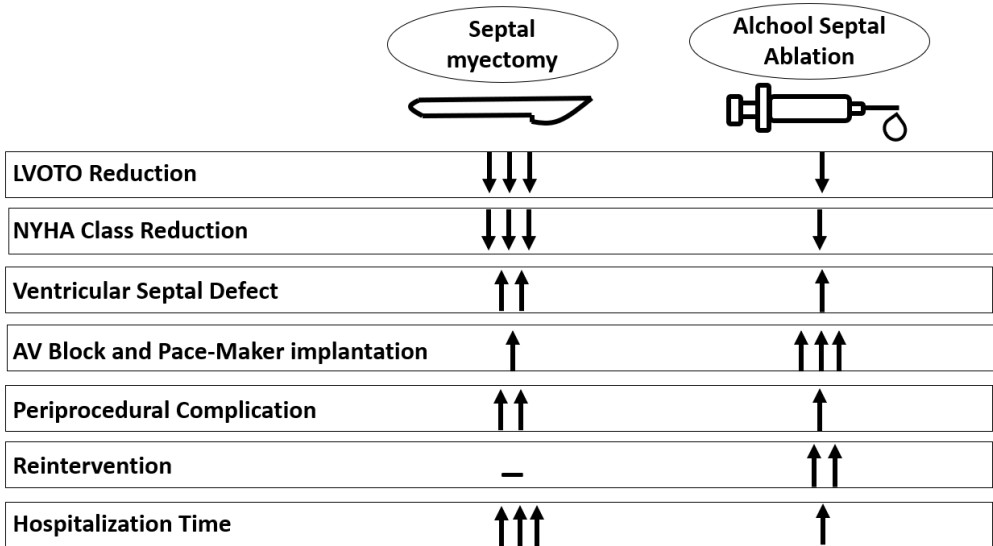

**Figure 14.** Comparison between septal myectomy versus alcohol septal ablation. AV = atrio-ventricular; LVOTO = left ventricular outflow obstruction.

## 5. LVOTO: Does It Really Exist?

In the late 1950s, the dynamic nature of muscular LVOTO, rather than valvular LCOTO, was elucidated by the utilization of bedside maneuvers. LVOTO was diagnosed if the patient was responsive to a manipulation of the LV load, in the absence of radiographic evidence of aortic valve calcification [47]. However, 10 years later, the existence of an impedance of LV ejection was disputed. Some researchers suggested that the difference in pressure between the LV cavity and the aorta was an artifact, caused by rapid ejection, complete systolic emptying and cavity obliteration [47]. The use of echocardiography from the late 1960s, demonstrating that LVOTO was present with an open rather than obliterated LV cavity, has softened this debate. In fact, during the echocardiographic evaluation of a HCM patient, cavity obliteration must be excluded, and SAM or MCO with an open LV cavity or apex must be demonstrated before determining that the pressure gradient represents true obstruction [47].

## 6. LVOTO in Other Conditions

LVOTO due to SAM was considered pathognomonic of HCM. However, the pathophysiology of LVOTO suggests that in some cases, LV morphological changes may lead to the development of an obstruction, as for example in the presence of hypovolemia, tachycardia, and/or reduced afterload. Dynamic intraventricular obstruction has been reported to occur in approximately 15% of patients undergoing aortic valve replacement or transaortic valve implantation (TAVI). These patients usually show a small LV diameter, asymmetrical hypertrophy, a high ejection fraction, and high valve gradients [48]. Systolic obstruction after aortic valve replacement or TAVI can be localized at the mid-LV region because in most cases aortic stenosis causes symmetric hypertrophy. However, sometimes in aortic stenosis, LV hypertrophy is asymmetric, which in some cases, together with a septal bulge, might induce LVOTO obstruction after aortic replacement or TAVI. The acute occurrence of severe LVOTO in these patients may provoke the so-called LV "collapse" or "suicide" [48]. Medical approaches to this emergency event include sufficient volume loading, titrated doses of beta-blockers to promote negative chronotropy and inotropy, and the administration of alpha-1 agonists to decrease the LVOTO gradient through increasing systemic vascular resistance. When medical therapy is unable to adequately relieve LVOT obstruction, surgical myectomy or alcohol septal ablation therapy and, as suggested in some case reports, the implantation of a pace-maker to induce septal desynchronization have been proposed [49].

SAM is not specific to HCM and may be present in several conditions. Surgical mitral valve repair can cause severe SAM; however, correction of hyperdynamic status usually resolves SAM and reduces MR in more than half of these patients. Patients with diabetes may manifest SAM in the context of LV hypertrophy due to a hyperdynamic state caused by increased β-adrenoreceptor sensitivity. Post-myocardial-infarction (MI) changes in LV geometry due to the presence of hyperkinetic and hypokinetic regions can sometimes result in SAM. General anesthesia, by causing vasodilation and hypovolemia, may provoke SAM even in the absence of any cardiac abnormality [49].

Future Directions. In HOCM, the invasive treatment of obstruction is becoming an increasingly distant option. Studies on the etiopathogenesis of HCM demonstrated that HCM often presents with early hypercontractility that stems from a high degree of actin–myosin cross-linking. Although the biomolecular cascade from the mutation to overt HCM is still not well understood, these studies have led to the development of new drugs such as mavacamten and aficamten, marking a new era in the treatment of LVOTO. In the future, gene therapy will most probably be the effective treatment for sarcomeric HCM with or without obstruction.

## 7. Conclusions

LVOTO is a common finding in HCM patients and may be present during rest or provoked by a Valsalva maneuver or exercise. Novel medical therapies seem to improve symptoms and outcomes and reduce the need for invasive procedures.

**Author Contributions:** Conceptualization, G.T. and M.A.L., writing—original draft preparation, G.T., Writing—review and editing, R.L., G.E. and M.A.L., supervision, M.A.L., formal analysis, G.C. and F.B., resources:, G.C., F.B. and E.F.P., data curation, G.C. and F.B. All authors have read and agreed to the published version of the manuscript.

**Funding:** PRIN-2017WACS from Ministry of University and Research; PNRR-MR1-2022-12376614 from the Italian Ministry of Heath (R.L.).

**Institutional Review Board Statement:** Not applicable.

**Informed Consent Statement:** Not applicable.

**Data Availability Statement:** No new data were created or analyzed in this study. Data sharing is not applicable to this article.

**Conflicts of Interest:** The authors declare no conflict of interest.

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
