# Peer review of "Diagnosis and Treatment of Obstructive Hypertrophic Cardiomyopathy"

_cardiogenetics, doi:10.3390/cardiogenetics13020008_

Round 1

Reviewer 1 Report

This is a review regarding obstructive HCM pathophysiology, diagnosis, and treatment. The authors should be proud of their work, as it is both extensive and pleasant to read.

Some minor reviews are needed:

1 - The echo pictures and PW/CW graphs seem quite old, and if possible should be updated to more current standard quality pics. 

2 - (mandatory for approval) A correction regarding the new drug name mavacamten should be performed (in table 1, there is a typo). Some english proofreading should be performed (see lines 228-230, p.e.)

3 - Since this is an in depth review, gender differences between HCM diagnosis and treatments (as different ICD implantation rates across the world, for instance) could be mentioned. 

Author Response

We thank the reviewers for his /her suggestions. We tried to answer to overall considerations, following them the review improved. R, refers to your suggestions; A: refers to our answers

R: This is a review regarding obstructive HCM pathophysiology, diagnosis, and treatment. The authors should be proud of their work, as it is both extensive and pleasant to read.

A: We thanks the reviewer for his/her consideration in our work.

R Some minor reviews are needed:

1 - The echo pictures and PW/CW graphs seem quite old, and if possible should be updated to more current standard quality pics.

A: A new picture has been added showing in the same frame the different shape of mitral regurgitation, intraventricular and outflow gradients.  

2 - (mandatory for approval) A correction regarding the new drug name mavacamten should be performed (in table 1, there is a typo). Some english proofreading should be performed (see lines 228-230, p.e.)

A: Thanks for your deeply attention in reading our paper. We corrected overall typos accordingly.

3 - Since this is an in depth review, gender differences between HCM diagnosis and treatments (as different ICD implantation rates across the world, for instance) could be mentioned.

A: we thank again the reviewer, in that we forgot to report gendered differences. We added a sentence in the Introduction section. “Genders differences have been reported in the role of LOVTO in the course of HCM. In fact, women showed greater likelihood of symptoms progression or death due to heart failure more associated than men with LVOTO []”, and we added the relative reference.”Olivotto I, Maron MS, Adabag AS, et al. Gender-related differences in the clinical presentation and outcome of hypertrophic cardiomyopathy. J Am Coll Cardiol. 2005;46:480-7”.

Reviewer 2 Report

Gaetano et al report about Diagnosis and treatment of obstructive hypertrophic cardiomyopathy.

Point: I am Radiologist with a speciality in cardiac imaging. 

The Work is a quick guideline, good, but it needs some changes.

1. line 74-76: Reference? This formulation leads to misunderstanding for readers. It may happen but not necessary. The authors need to clarify it.  

2. line 94: As a suggestion: a picture of Pseudo SAM makes the work very stronger, but not necessary.

3. fig 4 is in 2 chamber view and doesn't deliver midventricular obstruction.

4.  3.2. cardiac magnetic resonance: supplement CMR is very, very poor. Currently, CMR plays a very big role in the diagnostics HCM and HOCM, and authors should definitely expand and discuss imaging, flow measurement  and LGE and the role of LGE in prognosis

5. Differential diagnosis of HCM, especially in CMR and nuclear medicine, is absolutely missing. Discussion about amyloidosis, Fabry disease and etc, is a great weak point of the work and needs to discuss. 

Author Response

We thank the reviewers for his /her suggestions. We tried to answer to overall considerations, following them the review improved. R, refers to your suggestions; A: refers to our answers

R: Gaetano et al report about Diagnosis and treatment of obstructive hypertrophic cardiomyopathy.

The Work is a quick guideline, good, but it needs some changes.

A: we thanks the reviewer for his/her consideration in our work.

R 1. line 74-76: Reference? This formulation leads to misunderstanding for readers. It may happen but not necessary. The authors need to clarify it. 

A We added the relative reference” Cardim N, Galderisi M, Edvarsen T, et al. Role of multimodality cardiac imaging in the management of patients with hyper-trophic cardiomyopathy: an expert consensus of the European Association of cardiovascular imaging endorsed by the Saudi Heart Association. Eur Heart J CVI2015; 16:280”. In addition, we rephrased the sentence to better clarify the point of MCO and absence of obstruction “Patients with MCO and apical aneurysms may have not high mid-LV velocities because of the complete obstruction in the mid left ventricle, like a sphincter closing aneurysm. In addition, blood emerges paradoxically from the apex in diastole when the sphincter is released []”.

R 2. line 94: As a suggestion: a picture of Pseudo SAM makes the work very stronger, but not necessary.

R: thanks. We added a figure.

R 3. fig 4 is in 2 chamber view and doesn't deliver midventricular obstruction.

A: Thanks. We were wrong and we changed the figure with a correct one reporting a 3-chamber view with hourglass appearance. Now the figure is number 5.

R 4.  3.2. cardiac magnetic resonance: supplement CMR is very, very poor. Currently, CMR plays a very big role in the diagnostics HCM and HOCM, and authors should definitely expand and discuss imaging, flow measurement and LGE and the role of LGE in prognosis

R 5. Differential diagnosis of HCM, especially in CMR and nuclear medicine, is absolutely missing. Discussion about amyloidosis, Fabry disease and etc, is a great weak point of the work and needs to discuss.

A: Thank you very much. We added two figures and some sentences in the subheading CMR

Round 2

Reviewer 2 Report

In my opinion, the work has significantly improved and can be considered a quick guideline now.

Author Response

We revised the English language